# 'It's trying to manage the work': a qualitative evaluation of recruitment processes within a UK multicentre trial

Zoë Christina Skea, Shaun Treweek, Katie Gillies

Health Services Research Unit, Health Sciences Building, University of Aberdeen, Aberdeen, Scotland, UK

**Correspondence to**
Katie Gillies;
k.gillies@abdn.ac.uk

## ABSTRACT

**Objectives** To explore trial site staff's perceptions regarding barriers and facilitators to local recruitment.

**Design** Qualitative semi-structured interviews with a range of trial site staff from four trial sites in the UK. Interviews were analysed thematically to identify common themes across sites, barriers that could be addressed and facilitators that could be shared with other sites.

**Participants** 11 members of staff from four trial sites: clinical grant Co-applicant (n=1); Principal Investigators (n=3); Consultant Urologist (n=1); Research Nurses (n=5); Research Assistant (n=1).

**Setting** Embedded within an ongoing randomised controlled trial (the TISU trial). TISU is a UK multicentre trial comparing therapeutic interventions for ureteric stones.

**Results** Our study draws attention to the initial and ongoing burden of trial work that is involved throughout the duration of a clinical trial. In terms of building and sustaining a research culture, trial staff described the ongoing work of engagement that was required to ensure that clinical staff were both educated and motivated to help with the process of identifying and screening potential participants. Having adequate and sufficient organisational and staffing resources was highlighted as being a necessary prerequisite to successful recruitment both in terms of accessing potentially eligible patients and being able to maximise recruitment after patient identification. The nature of the research study design can also potentially generate challenging communicative work for recruiting staff which can prove particularly problematic.

**Conclusions** Our paper adds to existing research highlighting the importance of the hidden and complex work that is involved in clinical trial recruitment. Those designing and supporting the operationalisation of clinical trials must recognise and support the mitigation of this 'work'. While much of the work is likely to be contextually sensitive at the level of local sites and for individual trials, some aspects are ubiquitous issues for delivery of trials more generally.

**Trial registration number** ISRCTN No 92289221; Pre-results.

## INTRODUCTION

Recruiting the desired number of participants is crucial for all clinical research and remains a major challenge for those concerned with designing and supporting the operationalisation of clinical trials.[1] [2]

### Strengths and limitations of this study

► The approach of nesting qualitative research within the context of clinical trials is considered particularly useful for improving the evidence base for how we conduct clinical trials.

► Compared with the vast majority of studies focussing on clinical trials, participants' perspectives about why they do or do not choose to consent, this qualitative study focuses on the views and experiences of staff involved in the recruitment process.

► Our study highlights and draws attention to the initial and ongoing burden of trial work that is involved in various stages throughout the duration of a clinical trial—this notion and the burden and associated consequences it can place on trial sites has been somewhat buried in much of the previous literature.

► All the participants in our study were UK based and a self-selecting sample focussing on issues within one particular trial setting. However, we think our findings will be transferable to other clinical trial settings and contexts and offer important insights for those concerned with designing and supporting the operationalisation of clinical trials.

The vast majority of studies exploring issues around trial recruitment have focused on trial participants' perspectives and experiences particularly around why they do or do not choose to consent to participate in clinical trials.[3–7] Increasingly, researchers are turning their attention to investigating the views of staff directly involved in trial recruitment, recognising that they may offer valuable and important insights for improving recruitment and other trial processes.[8–10] Focussing specifically on barriers and facilitators to trial recruitment, these studies have tended to highlight the importance of issues such as building a research culture, ensuring adequate resources and the focus of the research for being of significance in terms of their potential to facilitate trial recruitment.[11–16] More recently, researchers have begun to explore and document the emotional impact or burden that trial

involvement can have on recruiting staff, an issue perhaps particularly pertinent in certain trial contexts (eg, cancer trials; trials involving children).[9 17] The few published studies that have attempted to explore this notion have drawn attention to the emotional challenges of dealing with issues relating to tensions between research and clinical roles; accepting the concept of equipoise; concerns about overburdening prospective participants and dealing with patients' disappointment with treatment allocation postrandomisation.[9 10 17 18]

Building on this notion of trial burden or trial 'work', in this paper we present data from a qualitative study conducted with trial site staff as part of an evaluation of recruitment processes within a UK multicentre trial (the TISU trial). The TISU trial is a UK multicentre trial comparing therapeutic interventions for ureteric stones. Urinary stone disease is very common with an estimated prevalence among the general population of 2%–3% (1.8 million people in the UK) and is a major burden on the National Health Service (NHS) resulting in over 84 323 finished consultant episodes and over 97 558 bed-days in England in 2011 – 2012.[19] Urinary tract stones are associated with severe pain as they pass through the urinary tract and can have a significant impact on patients' quality of life due to the detrimental effect on their ability to work and the potential need for hospitalisation. Between a fifth and a third of cases require an active intervention (stone removal) because of failure to pass the stone, continuing pain, infection or obstruction to urine drainage. The two standard active intervention options, used routinely in the NHS, are extracorporeal shockwave lithotripsy and ureteroscopic stone retrieval (via surgery).

As with many trials, recruitment within the TISU trial proved more difficult than had been anticipated at the outset and some sites did not meet their target expectations. The variability across sites relating to aspects of recruitment suggested nuanced differences in the processes relating to recruitment and we set out to explore trial site staff's perceptions regarding barriers and facilitators to local recruitment. For the purposes of the TISU trial, the aim was to identify trial-specific modifiable factors that could enhance the facilitators and remove the barriers to recruitment.

## METHODS
### Recruitment, sampling and consent
We adopted a pragmatic approach to address recruitment issues in the TISU trial by including four of eight participating UK trial sites in this qualitative study. We designed a sampling strategy that allowed for maximum variability within our sample, whereby trial sites were sampled using a positive and negative deviant approach, that is, 'good' and 'could do better' sites were identified and invited to participate. We applied this approach both in terms of considerations around the total number of participants recruited in each site and also in terms of number of participants screened compared with numbers converted

to recruits, that is, one site screened several participants and randomised a small proportion of these, whereas others screened much smaller numbers but randomised a larger proportion. Within these sites, we sought to interview a diverse sample including those who held a range of trial roles such as research assistant, research nurse, research data manager, study PI and those with various other clinical roles. Email information about the study was distributed by the TISU trial manager, which included an invitation to take part in an audio-recorded interview relating to issues around how easy, or difficult, it was perceived for sites to recruit trial participants. The study was approved by the North of Scotland Research Ethics Committee (13/NS/0002). All participants gave written consent before participating in a telephone interview.

### Data generation
One-to-one interviews were conducted by telephone between January and April 2015. They were semi-structured in nature (supported by a topic guide to ensure coverage of key issues) and conducted in a non-judgemental, conversational style by one of two interviewers (ZCS and KG) whom participants knew were non-clinical and not part of the team running the trial being discussed. Both interviewers were female and experienced qualitative/mixed methods researchers with a track record in methodological research in the context of clinical trials. One interviewer (KG) also held a previous role (6 years previously) as trial manager in the same clinical specialty (urology) but a different condition and intervention. Neither interviewer had a clinical background nor had they ever had a direct role in recruiting participants to clinical trials. Although this valuable prior experience and background helped to sensitise the study team to the potentially pertinent issues and areas that would be important to explore during interviews, KG was mindful not to disclose to participants her own assumptions and opinions in this regard to avoid influencing their responses.

### Data analysis
Interviews were audio-recorded and transcribed verbatim. Our approach to analysis was systematic and interpretive.[20] We adopted an interpretive approach in this study, both in terms of our chosen data collection methods (in-depth interviews) and approach to analysis (inductive thematic analysis using the Framework Approach). For this piece of applied research, we found the interconnected stages of the Framework approach particularly helpful in terms of their systematic and transparent nature—making the approach conducive for analysing data as part of a team. We familiarised ourselves with the whole data set and following initial familiarisation with transcripts, developed a thematic coding framework based on discussions about both a priori questions and issues identified as emerging from the data. Initial codes (text labels) from this framework were then systematically applied to the transcript data. Data management and

**Table 1** Sample characteristics

| Trial site | Number of interviewees | Job description | Recruiting performance |
|---|---|---|---|
| 1 | 5 | ► Grant Co-applicant<br>► Principal Investigator (PI)<br>► Consultant Urologist<br>► 2 x Research Nurses (RNa; RNb) | High screeners - low converter |
| 2 | 1 | ► RN | Low screener-high converter |
| 3 | 4 | ► PI<br>► 2 x RNa; RNb<br>► Research Assistant (RA) | High recruiting site |
| 4 | 1 | ► PI | Low recruiting site |

initial analytic coding was facilitated by the use of NVivo V.10 text software. NVivo V.10 text management software was used to mark specific pieces of data that were identified as corresponding to the thematic index codes. More generally, NVivo V.10 was also used to help organise the data to facilitate further analytic consideration and interpretation. The primary focus during the analysis was on the a priori study aims. Particular attention was paid to the types of judgement, beliefs and attitudes (including concerns) that people expressed in relation to recruitment processes within their particular trial site, including views about the barriers and facilitators affecting trial recruitment.

## RESULTS
### Sample size and characteristics
Twenty-five trial site staff were approached to secure 11 interviews which lasted between 34 and 62 min. Table 1 illustrates the spread of staff across the four selected trial sites. Staff described having a range of roles in the context of the TISU trial, from having been involved in protocol and outcome measurement development (eg, one participant was a coapplicant on the TISU grant application) through to having a direct role in the day-to-day processes involved in screening and consenting prospective trial participants.

When asked to describe what they thought had worked well or less well within their particular site, staff were able to identify a range of factors which they considered as having played a role in either positively or negatively impacting on their site's ability to function effectively and to access potentially eligible patients successfully and, where appropriate, convert these to recruited trial participants. It was evident from within our data that some of these factors could, to a large extent, be supported by or (in the case of identified barriers) mitigated by what we will describe as the initial and ongoing trial site 'work' engaged in by trial staff. This 'work' related to issues around ensuring engagement and 'buy in' to the trial from a range of clinical colleagues as well as work involved in managing organisational complexity and management of specific treatment preferences (held by both potential participants and other clinical colleagues). These are discussed in detail below.

### Trial work to access potentially eligible patients: *issues of engagement and planning for pragmatism*
#### Engagement
Site staff identified the requirement to work with clinical colleagues (as part of routine patient care), who were not directly related to the trial (but who were nevertheless important facilitators to recruitment) as having a potentially negative impact on the identification of eligible patients. These non-trial clinical colleagues were perceived as being hard to engage with about the trial (both at the outset of the trial and on an ongoing basis) for a variety of reasons. Reasons cited by trial staff included: demanding clinical workloads; high staff turnover (eg, junior doctors on rotation); along with clinicians who were perceived to be either not particularly interested in the specific clinical area that the trial was concerned with or simply not that interested in research in general:

*…to begin with, we did have some problems and they (clinical staff who help to identify patients) hadn't done lots of research before…So it was getting everybody to be trained with GCP; that took quite a bit of time … they hold up the study if we haven't got all those things in place… it hasn't been too bad other than keep educating the doctors …a new doctor will come along and then you've got to go through it all again…they only tend to be between three and 6 months and then you get another lot… when you're looking at a trial like TISU, which is going to running for quite a long time, you've got to keep that training and that enthusiasm going. Site 3 RNa*

*…senior colleagues who are not interested in pushing it forward (ureteric stone research), you can't change that…I think our recruitment, which would possibly triple if everybody was on board with it, should be, educate people with what they should be doing … Site 3 PI*

Trial staff discussed how this could be mitigated or potentially resolved by initial attempts to engage with and motivate clinical staff and also by ongoing and creative attempts to maintain trial 'visibility':

*I'll go down to the (department) at 9.00 when the doctors just walk in, just to make sure they've got their research heads on as well as their clinical heads and that they will ring us if there's a patient… So it's about making them think that research is a normal bit of the hospital, this is the norm as opposed to the exception… we went down… with information given to the registrars and consultants at our monthly meetings and there are posters on the wards. There is a file on every ward where they would get admitted…So getting those nurses engaged…… Yeah, cake usually works, doesn't it? Site 1 RNa*

Having a strong buy in from the site PI, particularly in terms of raising the profile of the study and generating support from other clinical colleagues was also discussed as having an important role in maintaining trial visibility and keeping external colleagues engaged:

*It's pretty much a top down process, I think, from the consultants and the registrars. Mr X obviously has an invested interest in the trial doing well and tries to make sure that the medical teams are aware of the trial and refers patients if at all possible… Site 1 RNb*

*He's (PI) since gone on sabbatical and what I've noticed is that actually our recruitment has taken a bit of a turn it's his trial, it's not anyone else's trial… I think that not having the PI around to kind of push things forward has made our recruitment take a little bit of a turn recently. it's [PI's] baby, you know, (laughter)… Site 3 RA*

### Planning for pragmatism

In addition to discussing the importance of engaging with and motivating clinical colleagues to ensure that all potentially eligible patients are identified, trial staff discussed various other challenges relating to their attempts to maximise the initial recruitment potential at their particular site. Some participants clearly felt their sites were better placed than perhaps others for this due to organisational arrangements such as, for example, having a large geographic catchment area (ie, having the potential for a large recruitment 'net' of eligible patients); patients being cared for on specific wards and therefore being easy to locate and patients being admitted to the site after a diagnosis (and therefore facilitating the streamlining of the initial identification process). However, all trial staff discussed the ongoing challenge inherent at any site of trying to ensure that all eligible trial participants were approached for potential participation in the trial. The work involved in this process was discussed including attempts to maximise existing resources in terms of ensuring good cross cover of staff as well as developing innovative and proactive recruitment strategies:

*we have a lot of databases which we kind of meticulously keep … we kind of really plan everything out …then I'll put notifications in (Research Nurse) calendar and it's just having that organisation down so we don't miss any. Site 3 RA*

*there's trials that we consider generally generic between us. So we cross cover as much as possible. TISU is one of those ones that is quite amenable really for cover, it's not too complex, it's not too difficult to get your head round … because of running (similar related trial) we were very familiar with pretty much all the aspects of it including the trial paperwork and the database… Site 1 RNb*

As can be seen in the above quotations, there was a general consensus that the perceived non-complex nature of this particular trial (as well as previous involvement in similar trials) could greatly facilitate the identification of patients and the cross cover process—and so make the 'work' easier. However, sufficient resources were recognised as necessary for both of these processes to happen and clearly some sites were in a better position than others in this respect with the 'work' of a trial often not being formally recognised by host institutions for those with clinical roles:

*we don't have a dedicated research nurse but when (RN) is around it works extremely well… the downside, when (RN) is away on leave, we don't have a dedicated staff do the leg work and get the patient across so there will be some week or two when we may not be able to contact patients… (if I could) have a dedicated research nurse capturing all the patients who are diagnosed …and bringing those to my attention so …we don't miss a single opportunity, that type of thing, it would be ideal.… This research activity is not linked into our job plans, we do it out of our goodwill. Site 4 PI*

*…for TISU I am quite reliant on doctors referring patients to me… for me is the biggest barrier to recruitment is not having the control over the patients who, or the way we can identify patients… I've had some lengthy discussions with Mr X about this and how we recruit patients. Site 2 RN*

### Trial work to maximise recruitment after patient identification: *managing organisational complexity and management of treatment preferences*
#### Managing organisational complexity

In addition to identifying a range of factors which could impact either positively or negatively on attempts to *access* potentially eligible patients at their particular site, trial staff reflected on both site and trial-specific opportunities and challenges which they believed could affect the success or otherwise of successfully recruiting identified patients. Perceived site-specific organisational arrangement opportunities included features such as having a dedicated research facility (which was regarded as facilitating the consent process by allowing more time for detailed discussions); having a dedicated trial operating list (which helped to reduce waiting times for the interventional procedures); performing one of the interventions at a connected private hospital (with the perception that patients were attracted to this) and waiting list 'incentives' , which resulted in trial interventions being offered significantly earlier than the same interventions out with the trial.

*The research centre I would say is definitely a benefit …the best way really is to bring them over …and spend time going through everything, and perhaps draw a few diagrams, explain the procedures to them. Also mop up any questions that they perhaps haven't understood from the doctors. Site 3 RNa*

*what has been quite nice about the trial, is that patients have to receive their treatment within 8 weeks of being randomised…from our side that's very different to what would happen normally. I mean, sometimes it's sixteen weeks, so I mean that's just for the (surgical intervention). For the [non-surgical intervention], that's done at a private hospital nearby… That is the main thing but for ethical reasons we try not to push that too hard as a reason for why people should do it. Site 3 RA*

Conversely, staff at other sites described not having a dedicated research operating list (although discussed their ongoing efforts in trying to secure one) and also discussed potential waiting list 'disincentives' at their particular hospital, whereby there was a significant disparity between waiting times for the two trial interventions (which they felt could work against them in terms of patients willingness to be randomised):

*Unfortunately, in this facility, sometimes the surgical option is not as quick as…the lithotripsy option. You can see that actually influencing people's decisions, which they can have the earliest. I mean we're really encouraged by the sort of a surgical facility to see them within the week, we're hoping that will improve things. But setting something up like a surgical facility, you don't do in a week or two. We've been waiting 5 weeks now to be assured that we've got the theatre staff to cover it. Site 1 RNa*

### Management of treatment preferences

In addition to various site-specific organisational opportunities or challenges which could clearly impact on trial recruitment and generate more or less work for trial recruiters, it was also apparent that various non-site specific trial factors could have an influence and these seemed to require ongoing work and negotiation on the part of recruiting staff. For example, this particular trial was comparing an invasive surgical procedure (which required a general anaesthetic) with a much less invasive one (although potentially more painful and requiring more hospital appointments). Staff described how it was common for prospective participants to express a preference for the latter procedure and described the ongoing communication strategies and efforts involved in trying to somehow find a 'balance' in their discussions with patients:

*A lot of patients seem a lot keener on the lithotripsy because obviously it's a much less invasive procedure… we do always say to them, "Well, look. If you have these…you can have up to three treatments … three treatments and if all three of those fail then you will end up having (surgery) anyway",*

*but generally patients are much keener to try the less invasive procedure first, which is understandable. Site 3 RA*

The condition this trial was addressing is a particularly painful, and often recurrent one and so the speed with which the pain could be alleviated was considered to have a major impact on both clinician opinions and patient decision-making. Furthermore, the patients approached for participation had often had prior experience of the condition and of the treatment options and so sometimes had formed quite rationale and personal preferences that were not simply based on misunderstandings:

*I know there's pros and cons to both treatments on this study so … if I feel the registrar has jumped in and decided the management plan without considering TISU… It is a terribly painful condition, ureteric colic with stones and I think the speed in with which you resolve it must have a major impact in both the medical staff impression on what to do for a patient as well as the patient's decision-making. Site 1 RNa*

*…one of the obstacles to recruitment is, that patients do express a preference for one treatment or the other based on their own circumstances… in a way I think that that's free choice…I think sometimes there's personal reasons that some people would prefer not to have a general anaesthetic, would prefer not to have to stay in overnight… Site 2 RN*

## DISCUSSION
### Principal findings

This study explored views and experiences of staff involved in recruiting to a trial investigating two methods for treating ureteric stones. The vast majority of studies exploring issues around trial recruitment have focused on trial participants' perspectives and experiences particularly around why they do or do not choose to consent. Perhaps as a consequence of this, recruitment interventions have tended to be targeted at the individual patient level.[8 21] In comparison, relatively few studies have focused on the views of individuals responsible for participant recruitment, particularly about the broader processes of recruiting participants to clinical trials.[8] Of those that have, issues such as building and supporting a research culture, ensuring adequate resources and the focus of the research have all been deemed as important in terms of their potential to facilitate trial recruitment.[11–16] Our study supports these findings and also highlights and draws attention to the important notion of the initial and ongoing burden of trial work that is involved in each of these aspects, building on emerging recent findings from studies which have attempted to uncover some of the emotional 'hidden challenges' staff can face with regard to recruitment.[9 18] In terms of building and sustaining a research culture, trial staff in our study described the initial and ongoing work of engagement that was required to ensure that clinical staff were both educated and motivated to help with the process of identifying and

screening potential prospective participants to the trial, which may have implications for all research studies run at such sites. Having adequate and sufficient organisational and staffing resources was highlighted as being a necessary prerequisite to successful recruitment both in terms of accessing potentially eligible patients as well being able to maximise recruitment after patient identification and having few or no dedicated research staff at a site clearly created extra work in terms of juggling research roles with other clinical duties and responsibilities. Furthermore, our study demonstrated that the nature of the research study design can potentially generate more challenging communicative work for recruiting staff which can prove particularly problematic. For example, having trial interventions that have very different waiting times can make it harder for recruiters to convince prospective participants that there be no disadvantage to them in being randomised (if, as is the case with the TISU trial, a desirable outcome is likely to be fastest route to alleviation of pain). Also, comparing invasive interventions (requiring surgery) to less invasive ones will always generate potential communicative work for recruiting staff presenting a trial to prospective participants who may tend to have a natural inclination to favour the less invasive procedure.

Our study findings resonate strongly with recent theoretical work designed with complex interventions in mind which has highlighted that researchers are less inclined to think through the various demands that trial processes can place on those engaged in recruitment.[22] The Normalisation Process Theory addresses issues important for successful implementation and integration of healthcare interventions into routine work and suggests that researchers could benefit from paying more attention to, for example, the varied contexts of trial sites and also considerations around how necessary trial processes 'fit' with routine practice.

### Strengths and limitations

The approach of nesting qualitative research within the context of clinical trials is considered particularly useful for improving the evidence base for how we conduct trials. Most qualitative research within this context to date has considered views about different trials in different centres with fewer studies exploring views about the same trial across different settings.[15] By exploring the views of staff working within the same trial but across different performing sites we were able to highlight a range of both generic and site-specific aspects that could impact on patient identification and recruitment—aspects that will likely be very much transferable to other trials in other settings and contexts.

Although our study was more concerned with improving understanding of complex issues relating to trial recruitment rather than generalisability of results, sampling considerations were nevertheless important. Rather than approaching the sampling of trial sites opportunistically, we instead set out to purposively select staff for our study based on considerations (although somewhat subjective ones) relating to site performance.[23]

However, as is the case with many qualitative studies, there was an *element* of convenience (or 'opportunistic') sampling in that ultimately we had no control over who agreed to be interviewed from our initial sampling 'framework' and it is important to reflect on how this might this have influenced our findings. We could only interview those staff who responded positively to our letters of invitation (11 out of 25 from the 4 out of 8 selected sites) and one can speculate that their views may have differed from non-responders and those from unselected sites. In terms of data saturation, we were satisfied that our sample size was appropriate and adequate in terms of enabling us to sufficiently answer our research aims.[24] We were also reassured that there was variation in perspectives and experiences within our sample from staff who held a range of roles and that our study supports and helpfully builds on various key findings from other related studies.[11–16]

### Practice implications

Recruiting to trials is a complex and ongoing process and is one, for trial site staff, that starts well before the process of attempting to consent a prospective participant (eg, during a one-to-one recruitment consultation). The initial and ongoing trial recruitment work necessary to support successful recruitment is critical and should be explicitly recognised in terms of influencing how successful a site will be in accessing all potential candidates and randomising them. Intervention developers concerned with improving recruitment rates could usefully focus on more than just the individual patient level factors that might be having an impact on this process. Equally, trial methodologists could potentially have some control over or at least be mindful of the various challenges at trial design/set up stage, as well as considering the likely advantages of selecting particular sites.[22] Although those tasked with designing trials will never be able to anticipate all the potential challenges that sites may face across the duration of their involvement in trial recruitment (this will likely vary both within and across different trial sites, although several factors are generic for all trials, for example, buy in from clinical colleagues not involved in the trial), they should be mindful that the range of practical and logistical challenges inherent in trial recruitment all require more or less work on the part of trial recruiters and so support needs to be both responsive and targeted.

By nature, some clinical interventions will be potentially more (or less) attractive to prospective participants, but there is often a real scientific need to evaluate them in trials with specific comparators. Rotation of junior clinical staff can be frequent and some clinicians may be more interested than others in research in general and/or the clinical focus of the trial which could impact trial recruitment over time. All these aspects, and the 'work' that is required to address any deficits, need regular reflection and monitoring. Previous research has highlighted the benefits of training for recruiting

staff who struggle with the concept of equipoise and who perhaps hold particular treatment preferences, which make communicating trial rationale to patients difficult.[25] Our study suggests that this training could perhaps be extended to offer communication training for staff faced with trying to recruit to studies with particular types of study designs (eg, those comparing certain interventions; those that have waiting time disparities, etc).

## CONCLUSIONS

Our paper has demonstrated and highlighted the hidden and complex 'work' that is involved in clinical trial recruitment. This notion of trial work and the burden, and associated consequences, that it can place on study sites has been somewhat buried in much of the previous trial literature. Those designing and supporting the operationalisation of clinical trials must recognise and support the mitigation of this 'work'. While much of the work is likely to be contextually sensitive at the level of local sites and for individual trials, some aspects are ubiquitous issues for delivery of trials more generally.

**Acknowledgements** The authors would like to thank all interviewees who agreed to take part in this study. The authors would like to thank the TISU Trial Group for their support with this project. In particular, the authors would like to thank Sarah Cameron (TISU Trial Manager) for her help in identifying and recruiting staff from the various TISU trial study sites.

**Contributors** KG conceived the study idea. KG designed the study and KG and ZCS developed the interview topic guide. KG applied for ethics approval and KG and ZCS collected the interview data. KG and ZCS conducted the data analysis and ZCS wrote the initial and subsequent manuscript drafts. All the authors contributed critically to discussions about interpretation of data and revisions of manuscript drafts. All the authors approved the final version.

**Funding** ZCS was supported by a core grant from the CSO (reference CZU/3/3) and a Wellcome Trust Institutional Strategic Support Fund award (reference RG12724-18). KG was supported by an MRC Methodology Research Fellowship (MR/L01193X/1). Transcription costs were supported by the National Institute for Health Research (NIHR), HTA programme (TISU project number 10/137/01). The views and opinions expressed therein are those of the authors and do not necessarily reflect those of the HTA programme, NIHR, National Health Service or the Department of Health.

**Competing interests** None declared.

**Patient consent** This study did not involve patients. We interviewed a range of NHS staff and obtained written consent prior in interview.

**Ethics approval** The study was approved by the North of Scotland Research Ethics Committee (13/NS/0002).

**Provenance and peer review** Not commissioned; externally peer reviewed.

**Data sharing statement** To access the remainder of the data, please submit a request to the authors and the Chief Investigator of TISU. The request should include full justification for why secondary analysis of this data is being sought.

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
