## [Reviewer comments · BMJ Open]

ARTICLE DETAILS

TITLE (PROVISIONAL)	"It's trying to manage the work": A qualitative evaluation of recruitment processes within a UK multi-centre trial.
AUTHORS	Skea, Zoë; Treweek, Shaun; Gillies, Katie

VERSION 1 - REVIEW

REVIEWER	Laura Sheard Bradford Institute for Health Research, UK.
REVIEW RETURNED	03-Mar-2017

GENERAL COMMENTS	I enjoyed reading this interesting paper about the various burdens and complexity of trial "work" undertaken in a UK multi-centre trial. I think it is definitely publishable but I have some suggestions for improvement: - The authors use the phrase "trial work" throughout the paper but do not explicitly define what this means. Having read the paper, I now understand it to mean a mixture of emotional labour and organisational complexity but unless a definition is given then this phrase may mean different things to different readers. I doubt the authors need to cite already published literature; a few sentences of their own definition of "trial work" will suffice.- I did not understand what the authors' relationship to the study/ study team was. It is stated in the Methods section that they were "not part of the team running the trial" but I would like some indication of how they became involved in their qualitative evaluation of it.- The sample size is low at 11 participants. I understand why the authors sampled the sites using the positive and negative deviant approach but I think there needs to be a statement to say why they chose not to expand out to all 8 sites as uptake for the interviews was not particularly forthcoming.- Throughout the Findings section, I thought there were too many verbatim quotations. For instance, on page 6 and at the top of page 7, there are three quotes each (six in total) to demonstrate two points. It would help with the clarity of argument if only the most relevant quotes were selected.- I thought the headings of the themes did not bring the content of them to the fore. The first theme is about ownership and engagement as an issue in trial recruitment. The second theme is about resources staff and equipoise. But these interesting thematic findings have been obscured by title headings about access and maximising recruitment. Therefore, it is difficult to pick out what the
---

	main inductive findings are.
--	------------------------------

REVIEWER	Guek Eng Lee National Cancer Center Singapore
REVIEW RETURNED	13-Mar-2017

GENERAL COMMENTS	The interview questions should be further assessed for any bias or whether they have been validated. There are no further discussions on major or minor themes, nor discussions on whether themes reached saturation.
--

REVIEWER	Adwoa Hughes-Molrey University of York, UK
REVIEW RETURNED	20-Mar-2017

GENERAL COMMENTS	Thank you for asking me to review this interesting manuscript, which is a qualitative evaluation of recruitment processes within a UK multi-centre trial. This is an interesting and well-written manuscript which will make a welcome addition to the literature. I have the following comments on the methods section which I hope the authors will find constructive. METHODS 1. Page 4, line 17: ‘...not part of the team running to the trial’ (‘to’ should not be present in this sentence) 2. I would like to see more detailed description of the methodology, in line with best practice guidelines for reporting qualitative research. a) It would be helpful to have greater clarity on the methodology adopted and the process by which the analysis was undertaken. b) I was not quite clear what the authors mean by the ‘approach to the analysis was systematic and interpretive’ (p 25)? Does this mean that the authors approached the work using interpretivism as the underlying paradigm? It would be helpful for the authors to clarify this. c) Whilst the authors state that thematic coding was used, it is unclear which approach this was derived from. For example, was this based on framework analysis, on interpretive phenomenological analysis, or on something else? From the reference cited I can deduce that the framework analytical approach was used; however this is not at all clear from reading the text. If framework was used, why was this chosen over a straightforward thematic analysis? 3. More information to demonstrate that reflexivity was seriously considered in the methods would be helpful too (other than the interviewees knowing the role of the interviewer). Such as the gender of the interviewer, their credentials and their experience and training, in line with COREQ. For example, had the interviewers had prior experience in similar roles as the interviewees? How would this have impact on their interpretation? The need to be reflexive and critical is particularly important when using the framework approach (which I assume was used).
---

VERSION 1 – AUTHOR RESPONSE

Reviewer: 1

Reviewer Name: Laura Sheard

Institution and Country: Bradford Institute for Health Research, UK.

Competing Interests: None

I enjoyed reading this interesting paper about the various burdens and complexity of trial “work” undertaken in a UK multi-centre trial. I think it is definitely publishable but I have some suggestions for improvement:

- The authors use the phrase “trial work” throughout the paper but do not explicitly define what this means. Having read the paper, I now understand it to mean a mixture of emotional labour and organisational complexity but unless a definition is given then this phrase may mean different things to different readers. I doubt the authors need to cite already published literature; a few sentences of their own definition of “trial work” will suffice.

Response: We have added a sentence of clarification re. trial work to page 5.

- I did not understand what the authors’ relationship to the study/ study team was. It is stated in the Methods section that they were “not part of the team running the trial” but I would like some indication of how they became involved in their qualitative evaluation of it.

Response: The authors work in a multi-disciplinary clinical trials unit (that includes trialists, non-trialists, social scientists etc) and engage in a programme of related methodological research (including qualitative evaluations). The authors were not actively involved in running the TISU trial (for example as trial managers, data co-ordinators or recruiters etc) but were rather involved as experienced qualitative researchers in this specific qualitative evaluation.

- The sample size is low at 11 participants. I understand why the authors sampled the sites using the positive and negative deviant approach but I think there needs to be a statement to say why they chose not to expand out to all 8 sites as uptake for the interviews was not particularly forthcoming.

Response: We did not feel the need to expand out to all sites as we were reassured that within our sample of 11 participants staff held a range of roles and varied in their perspectives and experiences. We were pragmatic and flexible in our approach and felt that that our sample size was appropriate and adequate in terms of enabling us to sufficiently answer our research aims. We were confident that key recurring issues relating to the burden of various aspects of trial work were apparent in our data and that no new themes were emerging in the latter interviews that would necessitate further exploration with subsequent additional interviews. We have added some clarification about this issue to page 10.

- Throughout the Findings section, I thought there were too many verbatim quotations. For instance, on page 6 and at the top of page 7, there are three quotes each (six in total) to demonstrate two points. It would help with the clarity of argument if only the most relevant quotes were selected.

Response: For clarity we have removed two quotes from pages 6/7

- I thought the headings of the themes did not bring the content of them to the fore. The first theme is about ownership and engagement as an issue in trial recruitment. The second theme is about resources staff and equipoise. But these interesting thematic findings have been obscured by title headings about access and maximising recruitment. Therefore, it is difficult to pick out what the main inductive findings are.

Response: In structuring the findings section of our paper we thought it useful to arrange our headings around the natural flow or structure of trial processes (i.e. from initial study set up, through identification of potentially eligible patients, and through to the process of recruitment) highlighting the nature of various elements of trial 'work' involved at each stage. However, we accept the reviewer's point that in doing so our headings perhaps slightly obscure or leave key themes within each stage somewhat buried. We like the reviewer's summary of the key themes associated with each stage and as such we have edited the headings to reflect this whilst amending slightly to correctly reflect some aspects of the data.

Reviewer: 2

Reviewer Name: Guek Eng Lee

Institution and Country: National Cancer Center Singapore Competing Interests: None declared

The interview questions should be further assessed for any bias or whether they have been validated. There are no further discussions on major or minor themes, nor discussions on whether themes reached saturation.

Response: Within qualitative research (as oppose to, for example, quantitative research concerned with the development of a structured questionnaire), in depth open-ended general style interview questions would not normally be 'validated' as described by reviewer 2. However, these types of questions would usually be developed after some team sensitivity to previous related literature, and also team discussion and reflection (both before and during data collection). In our case, our general interview questions were developed after a familiarisation with previous literature that has explored trial recruitment issues, and some study team discussion re. the content of the interview schedule (both before and during data collection). Re. data saturation – please see above related responses. As discussed above we have added some additional clarity around data saturation to page 10 and have attempted to highlight the main inductive findings in the findings section.

Reviewer: 3

Reviewer Name: Adwoa Hughes-Molrey

Institution and Country: University of York, UK Competing Interests: None declared.

Thank you for asking me to review this interesting manuscript, which is a qualitative evaluation of recruitment processes within a UK multi-centre trial. This is an interesting and well-written manuscript which will make a welcome addition to the literature. I have the following comments on the methods section which I hope the authors will find constructive.

METHODS

1. Page 4, line 17: '...not part of the team running to the trial' ('to' should not be present in this sentence)

Response: Thank you – this has now been edited.

2. I would like to see more detailed description of the methodology, in line with best practice guidelines for reporting qualitative research.

a) It would be helpful to have greater clarity on the methodology adopted and the process by which the analysis was undertaken.

b) I was not quite clear what the authors mean by the 'approach to the analysis was systematic and interpretive' (p 25)? Does this mean that the authors approached the work using interpretivism as the

underlying paradigm? It would be helpful for the authors to clarify this.

c) Whilst the authors state that thematic coding was used, it is unclear which approach this was derived from. For example, was this based on framework analysis, on interpretive phenomenological analysis, or on something else? From the reference cited I can deduce that the framework analytical approach was used; however this is not at all clear from reading the text. If framework was used, why was this chosen over a straightforward thematic analysis?

Response to 2 a-c above: We did indeed adopt an interpretive approach in this study, both in terms of our data collection methods (in-depth interviews) and approach to analysis (inductive thematic analysis using the Framework Approach). For this piece of applied research, we found the interconnected stages of the Framework approach particularly helpful in terms of their systematic and transparent nature - making the approach conducive for analysing data as part of a team. We have added some clarity to the methods section.

2. More information to demonstrate that reflexivity was seriously considered in the methods would be helpful too (other than the interviewees knowing the role of the interviewer). Such as the gender of the interviewer, their credentials and their experience and training, in line with COREQ. For example, had the interviewers had prior experience in similar roles as the interviewees? How would this have impact on their interpretation? The need to be reflexive and critical is particularly important when using the framework approach (which I assume was used).

Response: Both interviewers were female and experienced qualitative/mixed methods researchers with a track record in methodological research in the context of clinical trials. One Interviewer (KG) also held a previous role (6 yrs previously) as trial manager in the same clinical specialty (urology) but a different condition and intervention. Neither interviewer had a clinical background nor had they ever had a direct role in recruiting participants to clinical trials. . Although this valuable prior experience and background helped to sensitise the study team to the potentially pertinent issues and areas that would be important to explore during interviews, KG was mindful not to disclose to participants her own assumptions and opinions in this regard to avoid influencing their responses. We have added some more details to the methods section for clarification.

VERSION 2 – REVIEW

REVIEWER	Laura Sheard Bradford Institute for Health Research, UK
REVIEW RETURNED	04-May-2017

GENERAL COMMENTS	I am pleased with the all the authors' responses to my previous review. I would now recommend this paper for publication with no further revisions needed.
--